Site selection for subtropical thicket restoration: mapping cold-air pooling in the South African sub-escarpment lowlands

Duker Robbert
http://orcid.org/0000-0003-3514-2685 Cowling Richard M.
van der Vyver Marius L.
http://orcid.org/0000-0003-0919-7279 Potts Alastair J. potts.a@gmail.com
Botany Department, Nelson Mandela University , Port Elizabeth , South Africa
Livesley Stephen
Electronic publication date: 2020 Apr 23
Publication date: 2020
Volume: 8
Electronic Location ID: e8980
Received 2019 Sep 26; Accepted 2020 Mar 25
Copyright: © 2020 Duker et al.
Copyright year: 2020
Copyright holder: Duker et al.
License: This is an open access article distributed under the terms of the Creative Commons Attribution License, which permits unrestricted use, distribution, reproduction and adaptation in any medium and for any purpose provided that it is properly attributed. For attribution, the original author(s), title, publication source (PeerJ) and either DOI or URL of the article must be cited.
License URL: https://creativecommons.org/licenses/by/4.0/

Keywords: South Africa, Portulacaria afra, Frost, Thicket-Wide Plot experiment, Digital elevation model, Spekboom, Subtropical thicket, Nama-Karoo, Ecological restoration

Funding: National Research Foundation 95992 and 91452 Natural Resource Management Programmes Department of Environmental Affairs This work was supported by the National Research Foundation (Nos. 95992 and 91452) and the Natural Resource Management Programmes: Department of Environmental Affairs. The funders had no role in study design, data collection and analysis, decision to publish, or preparation of the manuscript.

==============================
Restoration of subtropical thicket in South Africa using the plant Portulacaria afra (an ecosystem engineer) has been hampered, in part, by selecting sites that are frost prone—this species is intolerant of frost. Identifying parts of the landscape that are exposed to frost is often challenging. Our aim is to calibrate an existing cold-air pooling (CAP) model to predict where frost is likely to occur in the valleys along the sub-escarpment lowlands (of South Africa) where thicket is dominant. We calibrated this model using two valleys that have been monitored during frost events. To test the calibrated CAP model, model predictions of frost-occurrence for six additional valleys were assessed using a qualitative visual comparison of existing treelines in six valleys—we observe a strong visual match between the predicted frost and frost-free zones with the subtropical thicket (frost-intolerant) and Nama-Karoo shrubland (frost-tolerant) treelines. In addition, we tested the model output using previously established transplant experiments; ∼300 plots planted with P. afra (known as the Thicket-Wide Plots) were established across the landscape—without consideration of frost—to assess the potential factors influencing the survival and growth of P. afra. Here we use a filtered subset of these plots (n = 70), and find that net primary production of P. afra was significantly lower in plots that the model predicted to be within the frost zone. We suggest using this calibrated CAP model as part of the site selection process when restoring subtropical thicket in sites that lie within valleys—avoiding frost zones will greatly increase the likelihood of restoration success.

Introduction

Frost can hamper the restoration of plant communities (Snowcroft & Jeffrey, 1999; Snowcroft et al., 2000; Curran, Reid & Skorik, 2010; Rorato et al., 2018). In this study, we calibrate a cold-air pooling (CAP) model to predict where frost may occur in the complex terrain of the sub-escarpment lowlands of South Africa where subtropical thicket is the dominant vegetation. We argue that mapping frost occurrence is crucial for thicket restoration, which primarily uses the frost-intolerant succulent shrub, Portulacaria afra (commonly known as ‘spekboom;’ Fig. 1A).

Figure 1 Portulacaria afra (spekboom), and the local- and regional treeline of thicket and Karoo shrubland.

(A) An approximately 10 year old individual of Portulacaria afra (spekboom) grown from a cutting that was planted as part of the Thicket-Wide Plot (TWP) experiment, (B) the local treeline of thicket and Nama-Karoo shrubland, and (C) the broader regional distribution of the subtropical thicket and Nama-Karoo shrubland biomes (DEM; Takaku, Tadono & Tsutsui, 2014; Vegetation; Dayaram et al., 2018).

Exposure to sub-zero temperatures can damage photosynthetic and metabolic activities in plants, and thereby reduce growth, reproduction, and/or survival (Osmond et al., 1987; Thomashow, 1999; Holdo, 2006; Körner, 2012a; Gusta & Wisniewski, 2013). Thus, the distribution of plant species is commonly influenced by the occurrence of sub-zero temperatures at a variety of scales, ranging from broad regions to the local landscapes (Rouse, 1984; Osmond et al., 1987; Körner, 2012b; Wakeling, Cramer & Bond, 2012; Duker et al., 2015a, 2015b; Muller, O’Connor & Henschel, 2016). For example alpine treelines—where tall woody vegetation is replaced by shorter shrubland—are vegetation boundaries driven by decreasing minimum temperatures associated with increasing elevation (Körner, 2012a). Frost has been suggested to be a driving determinant of the regional distribution of the Albany subtropical thicket (Potts et al., 2013), and—at a more local scale—of treelines between the Subtropical Thicket and Nama-Karoo biomes (hereafter referred to as ‘thicket’ and ‘Karoo shrubland’, respectively) (Duker et al., 2015a, 2015b). The frost-prone, continentally dry, and high-elevation interior plateau of South Africa is dominated by frost-tolerant Karoo shrubland vegetation with small patches of thicket vegetation found in frost-free refugia such as steep slopes and rocky outcrops (Hoare et al., 2006; Mucina et al., 2006; Fig. 1C). However, this pattern is reversed in the sub-escarpment lowlands, which are lower and warmer but are prone to CAP (hereafter CAP; Fig. 1B). These lowlands are topographically complex due to the Cape Fold Mountains trending from west to east, intersected by rivers running, roughly, from north to south. In the valleys, the Karoo shrubland only occurs in frost-exposed valley floor, surrounded by dense thicket in the neighbouring frost-free valley slopes (Duker et al., 2015a, 2015b). Frost occurs on the valley floor due to CAP (Schulze, 2007; Duker et al., 2015a, 2015b). When wind speeds are low or wind is absent, atmospheric buoyancy forces can drive the formation of steep temperature gradients—this is caused by denser cooler air decoupling from the free atmosphere and sinking below relatively more buoyant and warmer air (Goulden, Miller & Da Rocha, 2006; Lundquist, Pepin & Rochford, 2008; Dobrowski et al., 2009; Smith et al., 2010; Dobrowski, 2011). Such CAP in flat and low-lying areas is especially common in regions with complex terrain, where changes in elevation and catchments result in intense frost on valley floors while slopes remain frost-free (Barr & Orgill, 1989; Neff & King, 1989; Lindkvist, Gustavsson & Bogren, 2000; Chung et al., 2006; Bigg et al., 2014). This has led to the exclusion of taller, dense thicket vegetation from valley floors where one would expect it to be dominant on the deeper bottomland soils (Becker et al., 2015); this includes P. afra.

Portulacaria afra is an important species in the arid subtypes of the thicket biome (and where it is dominant, this is commonly referred to as “spekboom thicket”), and is considered an ecosystem engineer in this vegetation (Van der Vyver et al., 2013), where it grows in a tangle with numerous other shrubs, trees and creepers. Here, its dense growth form and high levels of litter production enriches the soil and provides shaded, relatively moist microsites for the germination and establishment of woody thicket canopy species (Sigwela et al., 2009; Wilman et al., 2014). This makes arid thicket particularly sensitive to disturbances such as livestock browsing, which selectively removes the highly palatable spekboom from the plant community, thereby changing the microclimate that is critical for persistence of other plant species. Historically high levels of livestock browsing—primarily by goats or sheep—have removed much of the spekboom in arid thicket. This has resulted in loss of ecosystem functioning and thicket biodiversity (Vlok, Euston-Brown & Cowling, 2003; Lechmere-Oertel, Kerley & Cowling, 2005; Lechmere-Oertel et al., 2008; Sigwela et al., 2009). Subsequent colonisation of species from the neighbouring Karoo shrubland is common (Hoffman & Cowling, 1990; Rutherford, Powrie & Husted, 2012). Recently, large-scale initiatives have begun using clonal propagation of P. afra in areas of livestock-degraded thicket, with the aim that this drought-hardy species will facilitate the restoration of thicket ecosystems. A problem is that many areas of degraded thicket are now physiognomically very similar to the Karoo shrubland, and this has meant that some restoration efforts have mistakenly focused on frost-exposed parts of the landscape, specifically valley floors. This significantly reduces survival and growth rates of spekboom (Van der Vyver, 2018). Thus, a means to select frost-free sites is required where thicket restoration efforts are being conducted in topographically complex landscapes.

To predict where frost events may occur in the landscape, we calibrated a CAP model (developed by Lundquist, Pepin & Rochford, 2008) for the subescarpment lowlands of South Africa; this model was developed for high elevations (>1,000 m) and latitudes (>37°) in the northern hemisphere. Here we calibrate the model to terrain at lower elevations (generally <1,000 m) and latitudes (~32–34°) in the southern hemisphere. The CAP model uses a digital elevation model to generate terrain characteristics (slope, relative elevation and curvature), and then threshold values are applied to each of these to classify whether a part of the landscape is prone to CAP. This model has been used to model CAP in other parts of the world (Curtis et al., 2014; Patsiou et al., 2017). An alternative means to model CAP is to use physics-based modelling that simulates mesoscale and finescale atmosphere flows. However, such high resolution simulations are computationally expensive and need to be adjusted to each location they are applied (Pagès, Pepin & Miró, 2017). Furthermore, such modelling is not necessarily superior in CAP prediction as they may not always predict the extent and severity of cold-air drainage (Pagès, Pepin & Miró, 2017).

We used two adjacent valleys in the southern sub-escarpment lowlands of South Africa that have been well-studied in terms of CAP to calibrate the Lundquist, Pepin & Rochford (2008) CAP model. We then tested the calibrated CAP model using two approaches: (1) by visually comparing the model predictions to the thicket–shrubland boundary in six valleys—as stated above, this boundary is similar to an alpine treeline, but inverted with trees above the CAP frost zone; and (2) data from the Thicket-Wide Plot (TWP) experiment (an extensive transplant experiment conducted across the thicket biome; further details provided in “Materials and Methods”) to determine if model successfully predicts the decline in P. afra growth rates that would be associated with CAP, and related frost occurrence, in mountain valleys. Note that this model predicts the CAP that occurs in complex terrain only (i.e. it cannot be used to predict CAP in flat terrain).

Materials and Methods

Unless otherwise specified, all analyses described below were conducted in R version 3.3.1 (R Core Team, 2016).

Region

The terrain of the coastal lowlands is particularly suited for producing CAP due to the characteristic valley morphologies. The west–east trending Cape Folded Belt consists of erosion resistant sandstones that form long parallel ranges with shales and small patches of Dwyka tillite persisting on the lower slopes and valley floors; this gives rise to steep convex slopes with wide flat valley bottoms (similar to the glacially carved U-shaped valleys in the Lundquist, Pepin & Rochford (2008) study). The long parallel ranges are each fairly narrow (rarely more than 10 km wide), separated by valleys that range in width (but usually not wider than 30 km). The ranges are cut through by very narrow defiles by rivers flowing from the Great Escarpment to the sea (i.e. roughly from north to south). The rugged ranges, open valley floors, with valley constrictions through mountains increase the potential for CAP in this landscape.

Our approach, in brief, was to calibrate the CAP model (Lundquist, Pepin & Rochford, 2008) using two adjacent valleys in the sub-escarpment lowlands where a range of CAP-related data have been collected (e.g. during and after frost events, including temperature, transplant experiments and field observations; Duker et al., 2015a, 2015b). Next, we tested the model predictions at six different localities—selected from satellite imagery (via Google Earth Pro)—using the visually distinctive treeline between the thicket and Karoo shrubland. Finally, we tested our CAP model by using productivity data of spekboom planted in the biome-wide thicket restoration experiment. We describe each of these steps in detail below.

Cold air pooling model calibration

Two valleys (S33°15′25.00″ E25°25′20.10″ and S33°15′10.34″ E25°24′13.34″; Figs. 2 and 3) were used for model calibration and frost-risk mapping. These sites were used because of the availability of 5 years of temperature data (from July 2012 to October 2017), and many hours of field observations during and after frost events (over 100 days spent on site spread across four consecutive winters from 2013 to 2016). These ‘calibration valleys’ are located in the Zuurberg mountain range in the north-eastern portion of the Cape Fold Mountains. Temperature loggers and field observations have confirmed that radiative frost (0 °C at ground level) occured on valley floors up to 15 times each winter as a result of CAP on still and clear nights (detailed in Duker et al. (2015a, 2015b)).

Figure 2 Distribution of CAP model calibration and testing valleys and TWPs in relation to vegetation and elevation.

Digital Elevation Model (DEM; Takaku, Tadono & Tsutsui, 2014) of the study region, the occurrence of spekboom dominated thicket (Vlok, Euston-Brown & Cowling, 2003), calibration and testing valleys, and the locations of each Thicket-Wide Plot (inside and outside CAPs) used in our study. Circles show location of the two calibration valleys. White-filled squares show location of the testing valleys. White-filled triangles show the location of the TWPs that were in CAP-prone areas. Black-filled triangles show the location of the TWPs that were in areas not prone to CAP.

Figure 3 Model calibration valleys in the Zuurberg Mountains.

The two frost model-calibration sites (Zuurberg Mountains) in the broader landscape (A) and in demonstrating the thicket-shrubland boundary in both valleys ((B) Buffels Nek; (C) Klipfontein; source Google, 2019, Maxar Technologies, CNES/Airbus, AfriGIS, Landsat/Copernicus). Red lines indicate the valley diameters used to calculate radii. Grassland and fynbos vegetation is found on hill crests due to a shift in geology. Points indicate the position of temperature loggers in the landscape.

The first valley (Klipfontein) is ~7.5 km long, ~3 km wide and ~300 m deep, whereas the second valley (Buffels Nek) is ~12 km long, ~2.5 km wide, and ~500 m deep (Figs. 3B and 3C). The vegetation patterns in these valleys consist of grassland and fynbos occupying the higher elevation sandstone ridges, and thicket dominating the shale-derived soils on the lower valley slopes, and Karoo shrubland occupying valley floors (which is a combination of shale- and alluvially-derived soils). In these valleys, soil conditions (i.e. particle size, depth, infiltration rate, sodium and electrical conductivity) are consistent across the valley floors and shale-derived sections of the valley slopes (Becker et al., 2015). Nonetheless, Karoo shrubland replaces thicket on the valley floor—this is largely driven by frost (Duker et al., 2015a, 2015b). In these valleys, some of the thicket-shrubland treelines are intact and clearly defined due to historically low levels of livestock pressure (Ian Ritchie, 2016, personal communication); this boundary has become blurred in places as livestock has removed thicket, and Karoo shrubland species have subsequently invaded. In such cases, frost-intolerant, but browser-resistant, indicator species (e.g. Pappea capensis) were used to identify the position of the treeline.

As outlined in the introduction, radiative frost on the southern African sub-escarpment lowlands predominantly occurs on valley floors and in depressions, and is associated with CAP. Following the Lundquist, Pepin & Rochford (2008) model, and using the ALOS 30 m (Takaku, Tadono & Tsutsui, 2014) digital elevation model (DEM), we generated the range of topographic characteristics—slope, height relative to surrounding cells and curvature—required to identify areas prone to CAP and radiative frost (Fig. 4). Specifically, the Lundquist, Pepin & Rochford (2008) model applies thresholds to maps of slope, rank elevation and curvature and combines these to generate a summary map predicting where CAP should occur. Slope was calculated using the Horn (1981) algorithm in the raster package (Hijmans & Van Etten, 2012; Fig. 4). Rank elevation for each DEM grid cell (i.e. percentile elevation relative to surrounding terrain) was determined by calculating it’s topographic position within a given radius in the valley landscape—the radius was manually calculated for each valley and is defined as half the distance between the two ridges that define the valley in which the DEM grid cell is situated (Fig. 4). Specifically, we calculated the ‘rank elevation of each DEM grid cell relative to the elevation of surrounding DEM grid cells within the square with the specified radius in each cardinal direction from the DEM grid cell’ (Lundquist, Pepin & Rochford, 2008). Calculating rank elevation in this way places each DEM grid cell within the context of the broader landscape (i.e. valley-scale), rather than the conventional approach that only uses the immediate neighbouring cells for rank calculations.

Figure 4 Surface maps of variables used to calculate CAP likelihood.

Maps of (A) elevation, (B) slope, (C) height relative to surroundings, (D) curvature, (E) CAP likelihood for the Klipfontein valley in the western portion of the Kaboega study site (Takaku, Tadono & Tsutsui, 2014).

The broad scale curvature was calculated using the Liston & Elder (2006) snow model formula: cv=14{12r(Z−Zw+Ze2+Z−Zn+Zs2)+122r(Z−Zsw+Zne2+Z−Znw+Zse2)}

where cv is the curvature at any particular DEM grid cell, Z is the elevation of that DEM grid cell, r is the relevant radius for the landscape under investigation, Zw/e/n/s is the elevation a distance r to the west/east/north/south of the DEM grid cell, and Zsw/ne/nw/se is the elevation a distance r to the south-west/north-east/north-west/south-east away from the DEM grid cell. Again, the curvature calculated here does not use the cells immediately surrounding each DEM grid cell, but is a derivative of the specific radius set by the user for the particular valley. Thus, this determines whether a DEM grid cell is on a ridge or within a valley within the broader surrounding landscape (Fig. 4). The curvature and rank elevation calculations used the same user-defined valley radius.

The Lundquist, Pepin & Rochford (2008) CAP model was developed in the northern hemisphere (beyond 37°N)—here we applied this same approach in the southern hemisphere, where continental temperatures are ameliorated by the larger ocean area and in a region that is at least 3° closer to the equator. The model thresholds used by Lundquist, Pepin & Rochford (2008) had poor results when applied to the sub-escarpment coastal plains in South Africa. Specifically, the model had a high degree of false positives—that is it predicted CAP far upslope beyond the thicket-shrubland tree-line. This is likely because the temperature gradients that form along these southern coastal lowlands are not as steep as those experienced at the high elevation (1,000–3,000 m) and latitude (37–40° N) study areas of Lundquist, Pepin & Rochford (2008). Therefore, we modified the threshold criteria based on observations and data of known frost events in the calibration valleys. Thus, our calibrated thresholds are to predict CAP are (i) slope less than 10° (originally: <30° in Lundquist, Pepin & Rochford (2008)), (ii) height relative to surrounding cells less than 30% (i.e. approximately the lower 1/3 of valleys; originally: <50%) and (iii) cells with curvature values less than 0 (unchanged). These modifications improved predictions, qualitatively assessed (i.e. a visual match up), of the thicket–shrubland treeline at the two calibration valleys. Most importantly, it was observed that frost only occurred in areas on the bottom of valleys, and in particular those with slope values less than 10° (the uppermost extent of the most extreme frosts). Visual matchups were conducted by generating surface maps of frost risk (a binary no risk or high risk), exporting these as a .kml file (kml package in R; Genolini et al., 2015) to Google Earth Pro, and comparing the CAP predictions with existing vegetation boundaries evident in the imagery (Fig. 5).

Figure 5 Results of CAP model at calibration valleys in the Zuurberg Mountains.

The naturally occurring biome boundaries between thicket and Karoo shrubland vegetation closely coincide with predictions of frost occurrence at the two calibration valleys ((A) Klipfontein; (B) Buffels Nek, source Google, 2020, Landsat/Copernicus, AfriGIS PTY (Ltd.) 2019). White lines indicate the position of the boundaries between thicket and Karoo shrubland vegetation.

Testing the calibrated CAP model

We used two approaches to test the regionally-adjusted CAP model. First, we assessed its accuracy in predicting the location of frost exposed areas in relation to thicket–shrubland treeline in other valleys. Second, we tested the model predictions for a wide range of valleys against the productivity of the frost-sensitive spekboom cuttings planted in the TWP experiment (described below).

Model prediction and treeline comparison

We assessed the model accuracy by visually by comparing the observed thicket-shrubland treeline and predicted CAP boundary in six additional valleys located in sub-escarpment lowlands of the Eastern Cape (Figs. S1–S6). In two of these valleys, the treeline was readily apparent (Figs. S1 and S2), whilst in the other four valleys the treeline had been disturbed by overbrowsing by domestic livestock (Figs. S3–S6). In these valleys the historical treeline was identified based on the visual persistence of clumps of thicket tree species (usually Pappea capensis, Searsia cf. longispina). Visual assessment followed the same process as described above (i.e. CAP predictions were exported to Google Earth Pro).

Model prediction and TWPs

Data from the TWP experiment were used to test the calibrated CAP model. The TWP experiment was established in 2005 through collaboration amongst numerous stakeholders including: the Subtropical Thicket Restoration Project, a community of scientists, government departments and thicket restoration practitioners. The primary aim of the experiment was to inform best-practice restoration protocols. Different planting techniques were tested to determine those that would maximise spekboom cutting survival and growth. Three hundred 50 × 50 m plots—fenced to exclude indigenous and domestic herbivores—were established in the degraded state of spekboom-rich thicket, mapped as Spekboom Thicket and Spekboomveld by Vlok, Euston-Brown & Cowling (2003). These plots spanned a wide variety of climatic and topographic environments (Fig. 2). During the establishment phase of this experiment, the potential of frost as a dominant driver of spekboom growth was not yet fully appreciated, especially by the implementers of the plots, who were warned to avoid bottomlands. Thus, the areas in which plots were established included valley floors—which may be frost-prone due to CAP. Because of this plots were positioned in landscapes in a climatically unbiased manner, and thus offer an opportunity to investigate the effects of frost on thicket restoration success.

In each of the 50 × 50 m plots, spekboom cuttings of various sizes were planted in different treatments to identify the most effective methods for maximising survival and phytomass production and ultimately stimulate thicket restoration (Van der Vyver, 2018). Here we use the phytomass production rates calculated from only one of the treatments (to avoid treatment-related biases)—specifically, cuttings of 50 cm in length and planted 15 cm deep as this had the greatest success rate. In brief, yearly rates of phytomass production for each plot was calculated via allometric sampling, i.e. non-destructive measurements of plant width and height are linked to overall biomass via an allometric model—this model was constructed by measuring plant characteristics, harvesting entire P. afra plants, and measuring the carbon content with conventional methods (see van der Vyver & Cowling, 2019, for details). Specific details on the calculation of phytomass production rate are provided in the Supplemental Information. Of the original 300 plots, only a subset were appropriate for testing the CAP model. A plot was excluded if: (1) it occurred outside of a valley (where mapping CAP would be inappropriate), (2) there was evidence of excessive herbivore browsing (by goats, sheep or Greater kudu) due to a lack of fence maintenance, and (3) it occurred at an elevation >1,200 m. A plot was considered to be outside of a valley if peaks or ridges could not be identified within 8 km of one another. We excluded plots above 1,200 m as our CAP model was calibrated for the sub-escarpment lowlands as we had existing data to conduct the calibration. At higher elevations, there are steeper temperature gradients and thus another calibration exercise would be required (with further data collection and field observations required).

Results and Discussion

To the authors’ knowledge, this is the first application of a CAP model in the southern hemisphere. The predictions of calibrated CAP model successfully matched the thicket-shrubland treeline in additional testing valleys and the phytomass production was significantly lower for TWPs found in the predicted CAP zone.

Vegetation boundaries in test valleys in the Eastern Cape

This regionally calibrated model correctly identified the thicket-shrubland treeline at five of the six test valleys (Figs. S1–S6). Where valleys were deeper relative to their width, and had a steeper change in gradient from the valley floors to valley slopes (i.e. Figs. S1–S6, excluding S3), the CAP (and thus frost presence) prediction aligned closely with the thicket–shrubland treeline. However, where the valley was shallower (relative to its width, that is Fig. S3), the model predicted a larger area covered by the CAP, and thus frost prone area, which extended ~50 m further upslope than the observed treeline between livestock degraded thicket and Karoo shrubland. Nonetheless, in such cases this minor overprediction (false positives) will still assist in landscape restoration planning to avoid planting in frost exposed areas. This predictive mapping of frost-risk (via cold-air pool mapping) may be considered somewhat simplistic as it is based on topographic features alone. Nonetheless, this approach does capture the areas in the landscape that would likely be prone to frost, especially in more mountainous or valley areas such as the Zuurberg mountain range. In addition, physics-based weather modelling has not been shown to have greater accuracy at predicting frost events (Pagès, Pepin & Miró, 2017).

In geomorphologically simpler terrain, such as the wide flatland areas found between the Great Escarpment and the Cape Fold Mountains, the CAP model as it is used here (in terms of the thresholds for slope and broad landscape position) would not be suitable for predicting the occurrence of frost. In these areas, where spekboom can be locally dominant, weather station data and field observations would need to be collected to establish the areas within the landscape that frost occurs, and its intensity under different weather conditions.

The TWP experiment

Of the 70 TWPs selected for the analysis, 30 plots occurred in frost-prone areas and 40 were in frost-free areas. Within this subset, spekboom planted in frost-free areas had significantly higher levels (72.2%) of mean aboveground phytomass production relative to those situated in the frost-prone areas (U = 720, p = 0.046; Fig. 6).

Figure 6 Aboveground phytomass production rates in the frost-exposed and frost-free TWPs.

Differences in plot-level mean aboveground phytomass production (t.C.ha−1.year−1) of spekboom cuttings in predicted frost-free and frost-prone areas in the 70 plots in the Thicket-Wide Plot (TWP) experiment (see the Supplemental Information for more details). Significant differences (α = 0.05) were determined using Mann–Whitney U-tests.

The great escarpment versus sub-escarpment coastal plains

The high elevation plateau of Southern Africa is a very different environment to the sub-escarpment coastal plains. The climate in the interior is more continental than that of the coastal plains, with corresponding lower mean and minimum temperatures and lower rainfall. We have observed that the topographic characteristics associated with frost on the high elevation plateau of South Africa are different to those of the sub-escarpment plains. For example frost occurs higher up, and on steeper slopes, in the Mountain Zebra National Park of the Eastern Cape than at the calibration valleys in the Zuurberg mountains and in more coastal areas, where it only occurs on very flat ground (Robbert Duker, 2013, personal observations). This Park is situated in the Sneeuberg Mountains on the Great Escarpment of South Africa, at much higher elevations (approximately 1,100–2,000 m) and with greater topographic heterogeneity (deeper and wider valleys) and structures (flatter mountaintops) than the calibration valleys in the Zuurberg mountains (400–900 m).

Frost in the interior thus seems to have greater intensity on valley floors where minimum temperatures can drop extremely low (<−10 °C), occurs much higher up valley slopes, and even occurs on the high elevation plateaux on mountain tops. In this area, the size of frost-free refugia where freezing-intolerant thicket species can survive are thus far smaller and more isolated here than they along the sub-escarpment, where the majority of the landscape is frost-free. Thus, frost characteristics differ substantially between the interior plateau and sub-escarpment lowlands. Thus, the CAP-model thresholds reported here that we used to predict frost occurrence in the sub-escarpment lowlands needs region-specific calibration (specifically slope and landscape-level topographic position).

CAP and climate change

At a regional scale, minimum temperatures and frost have been suggested to be important drivers of the distribution of thicket across glacial and interglacial conditions (Potts et al., 2013). However, at the landscape level, cold-air pooling is decoupled from regional climate change (Curtis et al., 2014), so it is challenging to predict how CAP-determined treelines will be influenced by warming average temperatures. Thus, the effects of sub-zero temperatures may still affect the distribution of thicket at a regional and elevational scale as fewer frost events occur, but treelines in CAP topography may or may not be affected. This requires monitoring, both in simple and complex terrain. We suspect that thicket treelines will be slow to respond to the increase in suitable areas for growth, as thicket regeneration dynamics are exceptionally slow (Cowling et al., 2005; Wilman et al., 2014).

Conclusions

A basic recalibration of Lundquist, Pepin & Rochford (2008) CAP model demonstrates that we can predict which areas are likely to be frost-prone. The reduced rates of aboveground phytomass production in the TWP plots that were situated in frost-prone areas compared to those in frost-free areas is a cause for major concern to restoration practitioners. Thus, we urge that this model should be used to identify no-go areas during the site-selection process for spekboom planting and thicket restoration.

Supplemental Information

Supplemental Information 1 Supplementary information.

Click here for additional data file.

Supplemental Information 2 Thicket-Wide Plot dataset used for testing of CAP model.

Click here for additional data file.

Andre Bezuidenhout and Ian and Ritchie (Kaboega Farming) are sincerely thanked for their provision of facilities, transport and accommodation that assisted with this study. We thank the three reviewers of this manuscript for their numerous questions and suggestions, which all substantially helped to improve the manuscript.

Additional Information and Declarations

Competing Interests

Author Contributions

Data Availability

Richard M. Cowling and Alastair J. Potts are Academic Editors for PeerJ.

Robbert Duker conceived and designed the experiments, performed the experiments, analysed the data, prepared figures and/or tables, authored or reviewed drafts of the paper, and approved the final draft.

Richard M. Cowling conceived and designed the experiments, authored or reviewed drafts of the paper, and approved the final draft.

Marius L. van der Vyver performed the experiments, authored or reviewed drafts of the paper, made available core data on the thicket-wide plots plus provided key insights into interpretation and usage of the data, and approved the final draft.

Alastair J. Potts conceived and designed the experiments, analysed the data, authored or reviewed drafts of the paper, and approved the final draft.

The following information was supplied regarding data availability:

This paper uses a freely available digital elevation model combined with analyses where the equations are reported in the article (Lundquist, J.D., Pepin, N. and Rochford, C. 2008. Automated algorithm for mapping regions of cold-air pooling in complex terrain. Journal of Geophysical Research 113 DOI 10.1029/2008JD009879). The aerial imagery used for testing the model is freely available at Google Earth. The data for the thicket wide plot comparisons are available as a Supplemental File.

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
