# Peer review of "Site selection for subtropical thicket restoration: mapping cold-air pooling in the South African sub-escarpment lowlands"

_PeerJ, doi:10.7717/peerj.8980_

## Round 0.1 · original submission · Major Revisions

Dear Dr Duker and co-authors,

Two reviewers found the manuscript well written, with good use of high-quality figures and tables for a study that is robust and suitably replicated. All reviewers could see that this could be a strong applied study of great assistance to those seeking to restore degraded patches of vegetation. However, there are several recommendations (some minor some major) that require attention.

In the introduction please include reference to those studies that have examined frost impacts in restoration efforts, before providing greater background detail on the Lundquist et al. (2008) model.

One Reviewer requests that you consider photographs as an addition – perhaps consider these as supplementary material?

Parts of the manuscript are imprecise and vague, and the blending of results with discussion masks the impact or obvious significant results/findings.

I strongly agree that you should restrict some of your varied terminology so that it is easier for the reader to spatially determine where they are in the study. R2 provides clear guidance to this issue. The limited use of terms means that they can be used as signposts through which the reader can navigate and move from one issue to the next. Use them wisely.

Please provide either hypotheses or very clearly stated research questions/objectives towards to the end of the introduction.
Please provide greater clarity on the experimental design. The large, empirical dataset provided by the ‘Thicket-Wide-Plot Experiment’ needs to be communicated more clearly.

Please concentrate on more clearly communicating the key results/findings before discussing them.

I look forward to receiving your revised manuscript in the near future, with a detailed response to all reviewers comments.

Regards

Steve

Reviewer 1 ·

Basic reporting

This manuscript is extremely well written, such that it was a joy to read, and the substance of the research was conveyed clearly. The introduction provided an excellent context to the study, though it would benefit from some mention of the few previous studies which examine the influence of frost in restoration.

The structure is also very good and the figures and tables are of high quality. I would suggest that the MS would benefit from additional use of photographs (see General Comments). The results are relevant to the hypotheses.

Experimental design

This MS is within the aims and scope of PeerJ. As described in the General Comments this research is of great importance and will make a very useful contribution to a clear (and substantial) knowledge gap.

The study is a robust one, drawing on several different approaches to address the aims. It is suitably replicated. The methods are described in appropriate detail.

Validity of the findings

The underlying data have been provided and represent a robust effort to address the aims. The conclusions are sound, relevant to the stated aims and are supported by the results.

Additional comments

This paper was a joy to read. It is well structured and well written, so it flowed very easily and was readily understood. It tackled an important and interesting topic in a robust way, producing sound findings that are not only interesting as regards a fundamental understanding of the mechanisms driving the boundary between Karoo and thicket biomes, but vital to practitioners seeking to restore degraded patches of thicket. Despite frost being a key determinant of vegetation patterns, scant research has been conducted on this environmental factor in the context of restoration ecology (for exceptions see Scowcroft & Jeffrey 1999; Scowcroft et al. 2000; Curran et al. 2010; Rorato et al. 2018). This paper will make an important contribution towards filling this large knowledge gap.

My main suggestion is that the manuscript could be improved by the addition of a photograph of the study species and by earlier reference in the main text to photographs showing the biome boundaries.

I have provided more detailed comments on an annotated version of the MS.

References

Curran, T.J., Reid, E.M. and Skorik, C. (2010) Effects of a severe frost on riparian rainforest restoration in the Australian wet tropics: foliage retention by species and the role of forest shelter. Restoration Ecology 18: 408-413

Rorato, D. G., Araujo, M. M., Tabaldi, L. A., Turchetto, F., Griebeler, A. M., Berghetti, Á. L. and Barbosa, F. M. (2018), Tolerance and resilience of forest species to frost in restoration planting in southern Brazil. Restoration Ecology 26: 537-542.

Scowcroft, P. G., and J. Jeffrey. (1999) Potential significance of frost, topographic relief, and Acacia koa stands to restoration of mesic Hawaiian forests on abandoned rangeland. Forest Ecology and Management 114:447–458.

Scowcroft, P. G., F. C. Meinzer, G. Goldstein, P. J. Melcher, and J. Jeffrey. (2000) Moderating night radiative cooling reduces frost damage to Metrosideros polymorpha seedlings used for forest restoration in Hawaii. Restoration Ecology 8:161–169.

Annotated reviews are not available for download in order to protect the identity of reviewers who chose to remain anonymous.

·

Basic reporting

Please see general comments below. While generally well-written, the language is scientifically imprecise and vague, hiding any potentially significant results.

Experimental design

See general comments below. In general, I found the experimental design to be unclear, as written. There is strength in the application and modification of the model to the Southern Hemisphere and the testing across a robust observational dataset. Unfortunately, the presentation of the work does not include hypotheses, and is unclear regarding description of the scale of the research (as currently written).

Validity of the findings

This could be the first application of the model to the southern hemisphere and restoration of thicket is a large concern for the region. I would be enthusiastic about efforts that can help guide management, especially to prioritize (spatially) where to do this restoration under limited resources. Some strengthening of this could be undertaken (see suggestions in general comments).

Additional comments

There are strong elements of this work that lend itself to eventual publication. This research appears to be the first test of a cold-air drainage model (frost prediction) in the Southern Hemisphere (but see comment below). This work also has potential for guiding restoration efforts in the Eastern Cape of South Africa, for a currently threatened and valuable ecosystem, thicket. The work leverages a broad-scale empirical dataset: the Thicket-Wide-Plot Experiment. One significant result of the study is the validation of lower productivity of spekboom inside frost areas vs. outside. Unfortunately, the paper suffers from several major limitations that preclude my support of publication at this time.

Major Comments

I believe the manuscript could be reworked to highlight some of the more significant findings. It seems to me that this is the first time the cold-air pooling model was applied in the Southern Hemisphere; this should be emphasized as a core outcome of the work.

There are no hypotheses associated with the study objectives. What did you expect to see, and why? Relate your expected results back to current scientific understanding and your specific results.

Overall, there were many places in the manuscript where the language could be more specific and tailored to the actual results of the study. For example, in the abstract, the authors refer to a “strong visual match.” Was this really a qualitative (and subjective) result, or one that was founded on cross-validation and statistical results (if the latter, report the details). Other examples include:
L 80 “functional ecosystems” – what does this mean in this context?
L105 “many hours of field observations” – how many?
L116 – “similar soil conditions” – explain what soil conditions you mean
L168 “resulted in improved predictions” – report the statistics

L173 “visual analysis” – what is this?
L181 – “boundary based test” – what is this?
I recommend the authors review the entire manuscript to avoid obscure or over-generalized language.

Even having gone through the paper several times, I found myself often confused about the scale (grain and extent) of the study, and how the data and sites mapped onto the methodological approaches. What I mean by this is that the authors seemed to interchangeably use the terms “along the sub-escarpment coastal plateau” (L 86), “site” (L95) or “valley” (e.g., L 111), a “grid cell” (L133) or “region of interest” (L160) or “broader surrounding landscape” (L 154), or “two valleys on the Koboega Farm, with only one GPS point provided). Similarly, the figures refer to the area covered by the WTP and the two valleys. By the end of all this, I was very confused!! I recommend the authors very clearly outline the grain/resolution of the study (for each question, if they differ). If there is a broader extent to which the study is relevant, this can be addressed in the intro and discussion. Throughout, consistent terminology would really assist the reader.

The Lundquist et al. (2008) model, given the fact that it is fundamental to this study, needs to be explained to the reader in the introduction. What did it do (and why) and WHY is this the best model to test in this region?

Throughout, I found that there was a mix of content among the sections (intro, methods, results, discussion). The authors should clearly map their text onto these sections. If it is appropriate for the journal to mix results and discussion (for example, the paragraph beginning L 248), fair enough. An unfortunate consequence of this choice, however, is that the results do not clearly stand-out, substantially diminishing the impact of the paper, unfortunately. Another example is L201-206 – This seems like an important motivator of the study and probably should go in introduction not in the methods.

The role of herbivory was unclear. Was this controlled for in this study? Was it a focus of the study? And/or is it simply a justification for the restoration goals that motivated the study? Moreover, it wasn’t clear if you were referred to livestock herbivory or natural herbivory. What is meant by “indigenous herbivory damage.” I wasn’t even sure if you were referring to humans or non-humans!

Minor comments:

Abstract:
-emphasize first-time application in S.hemisphere; emphasize the changes that you made to the model; emphasize the use of large-scale empirical dataset
-present actual results (with the most impactful details); highlight the key results!
-The final sentence says “cold air mapping should be conducted” – but isn’t this what you did? Perhaps make this last sentence more impactful for actual restoration efforts (presumably by managers who might not do all the work you did in the project).

Intro
L72 – from “forming”[?] clumps?
L72 – ‘causing breakdown of the microclimate’ – rephrase
L80 – drought hardy species (add hyphen for drought-hardy?)

Methods
L93 – Here is where I would have liked more introduction to the Lundquist model in the previous section
L93/94 – do you mean the empirical data (observed from TWP)?
In general – this first paragraph says what you did, but does not relate it to your questions (or hypotheses)
L107 – this is the first reference to “Cape Folded” and folks outside of SA will not know what you mean. Can you also provide a general climate description for the region?
L111 – are these the two valleys in the farm? Unclear.
L116 – remove (and elsewhere)
L121 – does “elsewhere” in this context mean in the study region, or outside of the study region? Clarify.
L123 – would be better to say what the adjustments were – the current subheading is uninformative (“adjustments to the Lundquist model”)
L130 – what does “rank” mean in this context?
L132 – what stats package did you use here? If this is R, you should provide the full R package citation here.
L142 – what is meant by “such as how a conventional slope or topographic position index might be calculated”? Seems like you are making assumptions here, but I am not sure what they are.
L146 – careful of the formatting of the equation – didn’t print well for me.
L159 – original used twice
L160 – “poor frost prediction” – provide details
L163 – what are the “high elevation” values in Lundquist?
L165 – what do you mean by “original” – in Lundquist, or in your methods?
L178 – “thicket shrubland boundaries at additional sites” – I have no idea, now, what is being tested or where
L180 – in general, you need to introduce the TWP in the introduction, particularly to mention that these are experimental plots. Also be consistent in using the acronym or not.
L181 – “the boundary-based test used three additional sites…” – now I am really, really confused about the scale of the study. Three additional plots?
L183 – “large areas” “many areas” – vague
L210 – why were only 70 of 310 plots useable?
L218 – the productivity data are not a key part of this paper as currently presented. In some ways that is fair, as you state in the introduction that you expected these results and that is in fact the motivating factor for the study (identifying frost-free zones). On the other hand, if the data is used to validate your model, it seems important to emphasize more. Right now, there is a mix of how these data are presented. The methods are in the supplemental, but the main text has the figure. Depending on how you want to frame these results, I would recommend consistency in how they are presented.
L243 – I don’t know what is meant by “purported boundary” here
L248 – much of these could be intro, too?
L274 – you’ve mentioned this experiment many times already, so why is it getting it’s own sub-heading now?
L295 –“very cold area’

Conclusions
Reframe this around your major results of validating a model in a new area and leveraging large experimental network. Make stronger case for restoration application and potential.

Reviewer 3 ·

Basic reporting

No comment.

Experimental design

No comment.

Validity of the findings

Line 240. It’s not clear where the boundary between thicket and shrubland is in Figure 4, so it’s difficult to fully assess this conclusion that the prediction was tightly aligned. I suggest another line is added to show where the boundary is.

Caption of Figure 4 mentions a red line. I couldn’t see it on the photos.

Additional comments

1. There are extensive plains around Jansenville, stretching towards Darlington Dam, that is known as Noorsveld. Various vegetation maps indicate that this huge area would have been dominated by spekboom. How does this fit in with the theorising on frost? It’s an area spanning more than 100 km in width, so of great importance from a restoration perspective. Do these plains experience frost? The introduction indicates that plains would generally not have been favoured by spekboom because of frost occurrence. These vast plains of Noorsveld seem to contradict that point. It would be useful for restoration practitioners to get the views of the authors set out in this paper. There are patches of this Noorsveld south of Jansenville (about 10 km South on the R75) where spekboom is indeed dominant. This is easily seen on Google Earth. This suggests that spekboom would have been a dominant plant in amongst the noors over enormous areas. Yet this Noorsveld presumably gets frost, being a plain?

2. Given that climate change is likely to increase average temperatures in the region, what is the likely effect on frost occurrence and thicket distribution? Climate change is an elephant in the room with regards to restoration, and it would be appropriate for the authors to address it. Perhaps frost occurrence will decrease and thicket will expand as a result? What are the authors views? It’s worthy of some speculation in the paper because the issue needs to be thought through and fully researched in the years ahead. It’s time to put it on the table and start dissecting it.

Line 126. Depressions, not depression.
Line 186. Semi-colon should replace the comma.
Line 195. The experiment, not this experiment. ‘This’ was used in the previous sentence. No need to repeat it.

---

## Round 0.2 · accepted · Accept

Dear Dr Duker and co-authors,
I am happy to accept your revised manuscript following a very positive review from one of the original reviewers.
Well done on a thorough and detailed revision and presentation of your work.
Regards
Steve

·

Basic reporting

The overall structure of the paper, particularly the framing of the study is much approved. I understand the authors' intention to not lead with hypotheses, which is certainly valid in many important scientific studies. That said, I believe the goals of the effort are now much more clearly identified and the authors' attention to this is much appreciated.

Experimental design

Much clearer explanation of the scale of the experimental design and the calibration/testing phases - thank you!

Validity of the findings

The visual comparison of the results is probably the weakest part of this evaluation - as there may have been rectification issues with the imagery, and/or the quantification of specific differences is impossible to infer at the site level. But, the authors interpret this approach fairly - only making generalizable conclusions across sites with similar characteristics.

Additional comments

I want to thank the authors for the careful revision of this manuscript. I was pleased to review it again and can confirm that my major concerns have been adequately addressed. The paper is methodologically much clearer and the clarity of the results is also vastly improved. Overall, this paper tests a cold-air-pooling model in two valleys in the Eastern Cape, South Africa and then compares these findings more broadly to a regional-scale data set. The results can aid restoration of thicket in this area by identifying areas that are prone to frost (and thus likely associated with lower spekboom productivity).